# Stigma and discrimination against transgender men in Bhutan

**Vinita Saxena**[1]*, **Audrey Xu**[2], **Kinley Kinley**[3], **Tashi Tsheten**[4], **Tenzin Gyeltshen**[5], **Tashi Tobgay**[6], **Tae Young Zajkowski**[7], **Willi McFarland**[8], **Lekey Khandu**[9]

**1** University of California Los Angeles, Los Angeles, CA, United States of America, **2** Stanford University, Stanford, CA, United States of America, **3** Department of Public Health, National HIV, AIDS and STIs Control Program, Ministry of Health, Thimphu, Bhutan, **4** Queer Voices of Bhutan, LGBTIQ Network of Bhutan, Thimphu, Bhutan, **5** Pride Bhutan, LGBTIQ Network of Bhutan, Thimphu Bhutan, **6** Institute of Health Partners, Thimphu, Bhutan, **7** University of California San Diego, La Jolla, CA, United States of America, **8** San Francisco Department of Public Health, San Francisco, CA, United States of America, **9** Department of Public Health, National HIV, AIDS and STIs Control Program, Thimphu, Bhutan

* vinsaxena01@gmail.com

## Abstract

### Background

While transgender people worldwide face high rates of stigma and discrimination, there are few studies of transgender men (also "trans men") in Asia. We measured the prevalence of, and factors associated with, stigma and discrimination faced by trans men in Bhutan to bring visibility to their experiences and inform health and social policy changes.

### Methods

This cross-sectional survey was conducted in nine regions in Bhutan from November 2019 to January 2020. A total of 124 trans men were recruited using a hybrid venue-based and peer-referral approach. Data were collected using an interviewer-administered question-naire. Multivariate logistic regression characterized associations with experiencing stigma and discrimination when accessing health services.

### Findings

Participants were young (48.0% 18–24 years) and 48.4% had migrated from a rural to an urban area. The majority (95.2%) experienced stigma because people knew or thought they were trans men. Associations with frequent experiences of stigma were living with their part-ner as a couple (adjusted odds ratio [AOR] 3.07, 95% CI 1.27–7.44) and being unemployed or a student (3.22, 1.44–7.19). Nearly half (47.6%) said they experienced discrimination when accessing health care because people knew or thought they were a trans man; this experience was associated with migration (2.42, 1.08–5.39) and having >15 trans men in their social network (3.73, 1.69–8.26). Most (94.4%) experienced verbal violence, 10.5% experienced physical violence, and 4.8% experienced sexual violence.

**Data Availability Statement:** Data cannot be shared publicly because we and our IRB have ethical concerns with releasing our study data publicly due to the small numbers of transgender persons in Bhutan, the high likelihood that sharing

the primary data may expose individual participants, and the possible harm that might befall participants if exposed. We appreciate that deidentification of a dataset usually means that participants cannot be identified individually. However, in the setting of our study and our sample size, even a deidentified dataset presents a high risk of revealing a participant's identity. Such data would effectively expose participants or could be perceived by the participants as being exposed. This type of scenario occurs in too many instances to safely make public the primary data. Requests for data will have to be considered on a case-by-case basis with extreme caution in what can be released to avoid exposing participants. Address requests for data access to the chairperson of the Research Ethics Board of Bhutan, Dr. Nezang Wangmo (or their successor) at: REBH, Ministry of Health, Royal Government of Bhutan, PO Box 726, Thimphu, Bhutan. Attention: REBH Secretary, Tel +975-2-322602, msgurung@health.gov.bt or tashidema@health.gov.bt.

**Funding:** This study was funded by the Royal Government of Bhutan Ministry of Health (REBH/Approval/2019/051), which supported LK, KK, TTb, TTs, and TG. Support for VS, AX, TYZ, and WM was provided by The National Institute of Mental Health (R25 MH119858). The funders had no role in study design, data collection, analysis, decision to publish, or preparation of the manuscript.

**Competing interests:** The authors have declared that no competing interests exist.

## Interpretation

Our study found high rates of stigma, discrimination, and interpersonal violence due to being a trans man in Bhutan. Findings highlight the urgent need for strengthening laws and regulations to protect the rights of transgender persons, particularly when accessing health services, recognizing partnerships, and preventing violence in public spaces.

## Introduction

Stigma, discrimination, and violence have been documented among sexual and gender minority people in many parts of the world, ranging from daily micro-aggressions to verbal abuse, physical assault, rape, and death [1]. Discrimination against transgender persons results in denial of academic and other economic opportunities [2]. Transgender men (also "trans men") are people whose sex is assigned female at birth but self-identify as men. Trans men also encounter abuse and harassment due to laws that aim to regulate public behavior and decency, including combating cross-dressing or imitation of the other sex [3].

Experiences of stigma, discrimination, and violence may prevent trans men's access to health and social welfare services. Compared to cisgender men, trans men in the United States are less likely to have health insurance and a usual source of health care [4]. Compared to cisgender men, they also experience a higher prevalence of health conditions such as emphysema, liver disease, ulcer, sleep disorders, and poor mental health [5]. The stigma and discrimination against transgender people can lead to hesitation in seeking health and social services, resulting in unmet physical and mental health needs. Unmet mental health needs in turn can lead to high rates of depression and suicide among trans men [6].

According to UNAIDS and the laws in Bhutan, there is no criminalization of transgender people, sex work, or same-sex sexual acts in private [7]. Bhutan's constitution protects transgender persons for acts such as rape, domestic violence, and sexual harassment in the workplace regardless of gender. However, laws do not explicitly protect LGBTQ+ individuals from discrimination on the basis of sexual orientation, gender identity, or gender expression. In addition, there are no legal explicit legal rights for transgender people to access gender-affirming care, marry, or change government identity documents. Pride Bhutan and Queer Voices of Bhutan, two civil society organizations advocating for LGBTQ+ rights, are not legally registered organizations [8]. It should be recognized that the legal and cultural situation in Bhutan is rapidly changing [9].

Unfortunately, there is very little scientific research published on trans men worldwide, and even less on trans men in Asia [10–13]. Further, to our knowledge, no studies in Asia have measured the prevalence of experiences of stigma, discrimination, and violence experienced by trans men. Therefore, we took the opportunity to ask questions in these domains in the first survey conducted for trans men in Bhutan.

The goals of the current report are to document the presence of stigma and discrimination experienced by trans men in Bhutan and identify factors associated with this stigma and discrimination. We further undertake to present the prevalence of different types of interpersonal violence experienced by trans men, including verbal, physical, and sexual violence. Such data can advocate for policies and practices to support transgender persons in Bhutan and establish a baseline from which to gauge future progress in eliminating stigma, discrimination, and violence toward this vulnerable population.

## Materials and methods

### Overall study design and source population

This study was a cross-sectional survey of trans men conducted in Bhutan from November 2019 to January 2020. Participants were eligible if they were 18 years or older and assigned female sex at birth and self-identified as a gender other than female regardless of hormone use, gender-affirming procedures, sexual orientation, and public gender presentation.

Study locations included nine of Bhutan's 20 districts (dzongkhag), namely Thimphu, Chhukha (Phuentsholing), Wangdue Phodrang, Sarpang, Paro, Samdrup-Jongkhar, Mongar, Punakha, and Bumthang. The study area covered 64% of the nation's population and 82% of the urban population. The largest and most urban districts are the capital Thimphu and Chhukha (Phuentsholing), the latter bordering India. These nine districts are located in the three major regions of Bhutan: central, southern border, and the east.

### Survey design and procedures

Details of the sampling and recruitment have been previously described [14] and are summarized here. First, community outreach workers recruited eligible trans men in their social networks to participate and instructed them to refer other eligible trans men from their networks. Second, peer outreach workers also recruited participants from online and physical spaces frequented by LGBTQ+ populations in Bhutan. This resulted in a study sample that was a hybrid of venue, online, and peer-referral recruitment methods. The dates when participants were enrolled are the same as the dates of the survey implementation. Eligible trans men who provided verbal informed consent participated in a structured face-to-face interview. The authors had no access to any information that could identify individuals during or after data collection, and the survey was anonymously conducted. As compensation, participants received cell phone airtime cards of ngultrum 500 (USD $6.79) for survey completion and ngultrum 200 (USD $2.72) for each peer referral.

### Measures

The structured questionnaire was brief and collected demographic information, use of services, and indicator questions on experiences of stigma and discrimination. These indicators are comparable with frameworks and definitions previously used for measuring stigma and discrimination among transgender populations [15]. Translated from the Dzongkha language, the question about stigma was generally asking about experience as a trans man, namely, "Have you experienced stigma because people knew or thought you are a trans man?" The question about discrimination referred to experience with accessing healthcare, namely, "Have you experienced discrimination when accessing health services because people knew or thought you are a trans man?" The answer choices for these two questions ranged from "often," "sometimes," and "never." Participants were also asked if they experienced violence (verbal, physical, sexual) due to being trans men. Occupations labeled as "entertainment" included employment in bars, karaoke clubs, and "drayangs" where dances are performed for customers. These occupations do not explicitly include sex work, although such venues are often locations where sex workers find clients [16]. The questionnaire also included questions on the level of outness about being transgender and the participants' social network size of other trans men.

### Statistical analysis

Descriptive analysis characterized the demographic profile of the study population and the prevalence of experiences of stigma, discrimination, and violence. We further examined

correlates of stigma and discrimination. For correlates of stigma experience, we dichotomized the outcome to be "often" vs. "sometimes/never" due to any experience reported by the vast majority of respondents (i.e., "often" or "sometimes" was reported by 95.2%). For correlates of discrimination when accessing health care, we dichotomized the outcome as "often/sometimes" vs. "never" due to a more balanced distribution of having experienced any such discrimination (i.e., "often" or "sometimes" was reported by 47.6%). We first performed bivariate analysis using the chi-square test to identify potential associations with experiences of stigma and discrimination. Multivariate logistic regression analysis was conducted to characterize independent associations with stigma and discrimination, considering the inclusion of variables associated with these outcomes at p<0.2 in bivariate analysis. Final models retained those variables with p<0.1, while considering only those with p<0.05 as significant. The final models are adjusted for the other variables included. Of note, age, education, and other demographic variables were not associated with stigma or discrimination, nor did they confound any of the associations with the variables included in the final model. Statistical analysis was conducted using StataSE 17.

### Ethics

This study was approved by the Research Ethics Board of Health (REBH) in Bhutan (Ref. No. REBH/Approval/2019/051). Oral consent was obtained from participants.

### Results

Table 1 provides descriptive statistics of this population. A total of 124 trans men were recruited. Participants were young, 48.0% aged 18–24 years, ranging from 18 to 37 years overall. All participants were assigned female sex at birth and currently identified as trans men. Historical terms for transgender persons in Dzongkha, the official language of Bhutan, were generally not used by participants of this study. Most participants (99.2%) were heterosexual, and all participants were born in Bhutan. The majority of participants (98.4%) had an education of middle secondary school and above, and 68.3% had an education level of high school and above. Among participants, 35.5% were living with a partner as if married, and 62.9% were single and never married. One-third (33.9%) held salaried jobs, one-third (33.9%) were students, 28.2% were unemployed, and 4.0% worked in entertainment jobs. Participants' current residences were 65.3% urban and 34.7% non-urban with nearly half (48.4%) having migrated from non-urban to urban areas.

Table 2 describes the stigma and discrimination experiences of trans men. The majority (85.5%) stated that many people know they are trans men. Concerning social networks, 51.2% knew more than 15 other trans men. More than half (58.9%) of participants reported often experiencing stigma due to being a trans man, while 3.2% reported never experiencing such stigma. In terms of experiencing discrimination when accessing health services because people thought or knew they were a trans man, 5.7% reported often, 41.9% sometimes, and 45.2% never experienced such discrimination. Most participants (94.4%) experienced verbal violence, 10.5% experienced physical violence, and 4.8% experienced sexual violence due to being a trans man.

Table 3 shows the prevalence of stigma and discrimination by different characteristics of trans men. Often experiencing trans-related stigma was significantly higher for individuals who were living with a partner as if married (77.3%, p = 0.002) compared to individuals who are single. By occupation, often experiencing stigma was higher among those who worked in entertainment (80.0%), were students (69.1%), or were unemployed (68.6%, p = 0.009). Often experiencing stigma was more frequent among trans men whose current residence was in an

**Table 1. Demographic characteristics of trans men in Bhutan, 2020 (N = 124).**

| Characteristics | N | % |
|---|---|---|
| Age group (years) | | |
| 18–24 | 59 | 48.0 |
| 25–37 | 64 | 52.0 |
| Sex assigned at birth | | |
| Male | 0 | 0 |
| Female | 124 | 100 |
| Intersex, other, don't know | 0 | 0 |
| Gender Identity | | |
| Male | 0 | 0 |
| Female | 0 | 0 |
| Trans woman | 0 | 0 |
| Trans man | 124 | 100 |
| Don't know, other | 0 | 0 |
| Sexual identity | | |
| Straight, heterosexual | 123 | 99.2 |
| Gay | 0 | 0 |
| Bisexual | 0 | 0 |
| Lesbian | 0 | 0 |
| Transmen | 1 | 0.8 |
| Other | 0 | 0 |
| Ethnicity | | |
| Bhutanese | 124 | 100 |
| Non-Bhutanese | 0 | 0 |
| Education | | |
| No education | 0 | 0 |
| Primary | 2 | 1.6 |
| Middle secondary school (grade 7–10) | 37 | 30.1 |
| Higher secondary school (grade 11–12) | 79 | 64.2 |
| University | 5 | 4.1 |
| Marital status | | |
| Living together, not officially married | 44 | 35.5 |
| Single never married | 78 | 62.9 |
| Divorced | 2 | 1.6 |
| Widowed | 0 | 0 |
| Other | 0 | 0 |
| Occupation | | |
| Salaried | 42 | 33.9 |
| Entertainment | 5 | 4.0 |
| Student | 42 | 33.9 |
| Unemployed | 35 | 28.2 |
| Region of birth | | |
| Urban | 23 | 18.6 |
| Non-urban | 101 | 81.5 |
| Current residence | | |
| Urban | 81 | 65.3 |
| Non-urban | 43 | 34.7 |
| Migration | | |

(*Continued*)

**Table 1.** (Continued)

| Characteristics | N | % |
|---|---|---|
| Non-urban to urban | 60 | 48.4 |
| Urban to non-urban | 2 | 1.6 |
| No migration | 62 | 50.0 |

*Categories do not always add up to the total due to missing data; percentages shown are among respondents.

urban area (66.7%, p = 0.015). Experiencing any discrimination was higher among trans men who were living with a partner (65.9%, p = 0.002), with current urban residence (55.6%, p = 0.015), who migrated (61.3%, p = 0.002), and who knew more than 15 trans men (65.1%, p<0.001).

Table 4 shows independent associations for often experiencing stigma and experiencing any discrimination among trans men in Bhutan. Those who were living with a partner as married had 3.07 times the odds of often experiencing stigma as compared to those who did not live with a partner (95% CI 1.27–7.44). Those who were students or unemployed had 3.22 times the odds of often experiencing stigma as compared to those with paying jobs (95% CI 1.44–7.19). Trans men who had migrated experienced 2.42 times the odds of experiencing any

**Table 2. Stigma and discrimination experiences of trans men in Bhutan, 2020 (N = 124).**

| Question | N* | % |
|---|---|---|
| Do people know you are a trans man? | | |
| No one knows this about me | 1 | 0.8 |
| Only a few friends/families know | 15 | 12.1 |
| Many people know | 106 | 85.5 |
| Don't know | 2 | 1.6 |
| How many other trans men do you know? | | |
| 1–15 | 60 | 48.8 |
| >15 | 63 | 51.2 |
| Have you experienced stigma because people knew or thought you are a trans man? | | |
| Often | 73 | 58.9 |
| Sometimes | 45 | 36.3 |
| Never | 4 | 3.2 |
| Don't know | 2 | 1.6 |
| Have you experienced discrimination when accessing health services because people knew or thought you are a trans man? | | |
| Often | 7 | 5.7 |
| Sometimes | 52 | 41.9 |
| Never | 56 | 45.2 |
| Don't know | 9 | 7.3 |
| Have you experienced violence because people knew or thought you are a trans man? (Multiple responses allowed) | | |
| None | 7 | 5.6 |
| Verbal | 117 | 94.4 |
| Physical | 13 | 10.5 |
| Sexual | 6 | 4.8 |

*Categories do not always add up to the total due to missing data; percentages shown are among respondents.

**Table 3. Associations with stigma and discrimination among trans men in Bhutan, 2020 (N = 124).**

| Variables | Often experienced stigma, N = 73 (58.9%) | p-value | Any discrimination in accessing health services, N = 59 (47.6%) | p-value |
|---|---|---|---|---|
| Age group (years) | | 0.353 | | 0.948 |
| 18–24 | 32 (54.2) | | 28 (47.5) | |
| 25–37 | 40 (62.5) | | 30 (46.9) | |
| Education | | 0.472 | | 0.354 |
| Middle secondary and below | 21 (53.9) | | 16 (41.0) | |
| Secondary school and above | 51 (60.7) | | 42 (50.0) | |
| Marital Status | | 0.002 | | 0.002 |
| Living together, not officially married | 34 (77.3) | | 29 (65.9) | |
| Single (includes divorced) | 39 (48.8) | | 30 (37.5) | |
| Occupation | | 0.009 | | 0.263 |
| Salaried | 16 (38.1) | | 15 (35.7) | |
| Entertainment | 4 (80.0) | | 2 (40.0) | |
| Student | 29 (69.1) | | 23 (54.8) | |
| Unemployed | 24 (68.6) | | 19 (54.3) | |
| Region of birth | | 0.104 | | 0.369 |
| Urban | 17 (73.9) | | 9 (39.1) | |
| Non-urban | 56 (55.5) | | 50 (49.5) | |
| Current residence | | 0.015 | | 0.015 |
| Urban | 54 (66.7) | | 45 (55.6) | |
| Non-urban | 19 (44.2) | | 14 (32.6) | |
| Migration | | 0.362 | | 0.002 |
| Urban/non-urban migration | 39 (62.9) | | 38 (61.3) | |
| No migration | 34 (54.8) | | 21 (33.9) | |
| Do people know you are a trans man? | | 0.191 | | 0.172 |
| No one knows this about me | 0 (0.0) | | 0 (0.0) | |
| Only a few friends/families know | 8 (53.3) | | 10 (66.7) | |
| Many people know | 65 (61.3) | | 49 (46.2) | |
| How many other trans men do you know? | | 0.338 | | |
| 1–15 | 33 (55.0) | | 18 (30.0) | <0.001 |
| >15 | 40 (63.5) | | 41 (65.1) | |

*Categories do not always add up to the total due to missing data, percentages are among respondents.

**Table 4. Multivariate analysis: Independent associations with stigma and discrimination experiences of trans men in Bhutan, 2020 (N = 124).**

| Outcome | Predictors | Adjusted* odds ratio (95% confidence interval) | p-value |
|---|---|---|---|
| **Often experienced stigma** | Living with partner | 3.07 (1.27–7.44) | 0.013 |
| | Student/unemployed | 3.22 (1.44–7.19) | 0.004 |
| | Current urban residence | 2.19 (0.96–4.99) | 0.064 |
| **Experienced any discrimination in accessing healthcare** | Living with partner | 2.11 (0.91–4.89) | 0.081 |
| | Any urban/non-urban migration | 2.42 (1.08–5.39) | 0.031 |
| | Trans men's social network >15 | 3.73 (1.69–8.26) | 0.001 |

*Adjusted for the other variables listed in each model.

discrimination in accessing health care compared to those who had not migrated (95% CI 1.08–5.39). Those who knew more than 15 trans men had 3.73 times the odds of experiencing any discrimination when accessing health services compared to those who knew fewer trans men (95% CI 1.69–8.26).

## Discussion

Our study measured very high levels of stigma, discrimination, and violence due to being trans male in Bhutan. Our data corroborate high levels of stigma and discrimination suggested by the few available studies of trans men worldwide [10–13,17]. Many of these studies were qualitative and therefore direct comparison to the prevalence found in our data is not possible. Nonetheless, a consistent pattern of high prevalence of stigma and discrimination is evident. For example, experiences of stigma were common in a study of trans men in Puerto Rico, with components described as structural, interpersonal, and individual [10]. Similarly, in Nigeria, trans men expressed frequent stigma when accessing HIV services for being transgender [11]. In this Nigerian trans male population, respondents specifically mentioned struggling with gender identity disclosure, facing reduced sensitivity when interacting with healthcare providers, and not receiving gender-inclusive HIV services [11]. Participants in a study with trans men in Uganda reported stigma related to disclosing their gender identity and HIV status, including at healthcare facilities, leading to decreased access to services [12]. A study in the United States found that trans men were less likely to seek medical care due to discrimination [13]. At least one quantitative study, from Peru, reported higher levels of discrimination experienced by trans men from healthcare providers (69.0%) than we found in Bhutan (47.6%) [17]. To our knowledge, our study is the first to measure stigma and discrimination among trans men in Asia. Also, to our knowledge, it is the largest study of trans men in Asia, with 124 participants.

Although the sample size was small in absolute terms, we were able to identify significant correlates of stigma and discrimination. For example, living with a partner was associated with higher odds of experiencing stigma and discrimination. As same-sex marriage is not yet officially recognized in Bhutan, marriage for transgender persons remains unclear. We therefore take the response of "living together" when asked about their marital status as a proxy for being married. We hypothesize that the association with stigma and discrimination may be due to the increased visibility a co-habiting couple may face in public. Current urban residence was also associated with experiencing higher levels of stigma and migrating to an urban area was associated with increased discrimination. We hypothesize a similar reason may explain these findings in that trans men in an urban area may be living more openly as transgender and therefore more easily targeted by stigma and discrimination. The finding that having more trans men in their social network being associated with increased discrimination may also support this increased visibility hypothesis; that is, being part of the community may entail higher visibility as transgender. Moreover, the questions themselves were phrased with the implication that the experienced occurred because people knew or thought they were trans men. However, because this study is cross-sectional, there may be a reversal of this cause and effect. That is, trans men experiencing stigma and discrimination may move to urban areas for greater anonymity, and they may live with their partner and seek out other trans men for increased safety or support after having faced stigma and discrimination. Our finding that trans men who were students or unemployed did not report higher levels of discrimination was unexpected because discrimination from academic and employment opportunities have been noted in previous studies of transgender persons [18,19].

Experiencing stigma and discrimination due to being trans men can also cause multiple adverse mental health effects. One study found that interpersonal stigma, which includes

interpersonal violence and everyday stigma, was associated with poor mental health [20]. These researchers also found that experiencing everyday discrimination was associated with depression, anxiety, post-traumatic stress disorder, and self-injury in trans men. A Canadian study that investigated the mental health of transgender youth, including transmasculine youth, posited that high rates of mental health problems might be attributed to stigma and related social and familial struggles [21]. Another study in New Zealand found trans men had higher prevalence of anxiety and depression diagnoses than trans women [22]. The researchers attributed the findings to trans men experiencing additional stressors, including higher reported rates of sexual abuse, domestic violence, and discrimination when accessing health-care and employment services. Trans men might also use substances as a way of coping with the stigma and discrimination they experience. One study found that 27.6% of trans men reported substance use as a coping technique for experiencing stigma [23]. Experiences of stigma and discrimination by transgender persons could also lead to eating disorders. A study of sexual and gender minority populations found transgender and gender non-conforming adults more likely to report current or past eating disorders compared to gay men [24]. Of note, other evidence finds trans men with protective factors to potentially mitigate mental health disorders. Such protective factors for trans men are being in a committed relationship, older age, increased resilience, and family connectedness [20–21]. Fostering these protective factors could provide increased support and benefit to trans men facing multiple levels of stigma and discrimination.

This study has limitations that must be noted. First, as mentioned above, the cross-sectional design does not allow for determining the temporal cause-and-effect relationships between cor-relations. Second, because the survey was brief, it included only a few questions on experiences of stigma and discrimination providing limited context and detail. We acknowledge the need for primary theoretical research on the context and measurement of stigma and discrimination for sexual and gender minority populations in Bhutan. Third, the representativeness of our sample is not known. We note that the young age of the sample is consistent with Bhutan's age structure [25]. However, the proportions in our sample that are urban and with secondary edu-cation appear higher than among the general population. Whether this is true of all trans men or only for those whom we could reach and who were willing to participate in our study is not known. We highlight factors of our study that may strengthen inference. Recruitment was not conducted through clinics or services, as is often the case for studies of trans men [22,26,27]. Moreover, recruitment included hybrid methods that helped further diversify the sample by reaching trans men who are less visible (i.e., through peer referral) as well as those who may be less connected to peers (i.e., through online and physical venues). Although the sample size is small, 124 participants, it should be noted that this is estimated to constitute a large proportion of the reachable transgender population estimated at 0.06% of adult men in Bhutan [14]. Finally, we also note that our survey is one of the largest of trans men to date. We know of only one online-recruited survey of greater sample size [28] and one other peer-referral plus venue-based recruited survey of equal size [29], both predominantly of trans men living in the United States.

In summary, our data point to very high rates of stigma and discrimination experienced by trans men in Bhutan, in line with many other parts of the world [10–13]. These experiences appear associated with factors that increase their visibility in public where they may face multi-ple contexts for stigma and discrimination, including when seeking health services. Our data speak to the need to strengthen anti-discrimination legislation to include gender identity and sexual orientation and recognize committed partnerships. While dismaying to see such high levels of physical and emotional violence directed at this community, we believe documenta-tion of their high prevalence can help advocate for these policies and laws to protect Bhutanese trans men to make their lives healthier and happier.

## Author Contributions

**Conceptualization:** Vinita Saxena, Audrey Xu, Tae Young Zajkowski, Willi McFarland, Lekey Khandu.

**Data curation:** Vinita Saxena, Kinley Kinley, Willi McFarland.

**Formal analysis:** Vinita Saxena, Tae Young Zajkowski, Willi McFarland, Lekey Khandu.

**Funding acquisition:** Kinley Kinley, Tashi Tobgay, Willi McFarland, Lekey Khandu.

**Investigation:** Kinley Kinley, Tashi Tsheten, Tenzin Gyeltshen, Willi McFarland, Lekey Khandu.

**Methodology:** Kinley Kinley, Tashi Tsheten, Tenzin Gyeltshen, Tashi Tobgay, Willi McFarland.

**Project administration:** Tashi Tsheten, Tenzin Gyeltshen, Tashi Tobgay, Lekey Khandu.

**Resources:** Tashi Tsheten, Tenzin Gyeltshen, Tashi Tobgay, Lekey Khandu.

**Supervision:** Tashi Tsheten, Tenzin Gyeltshen, Willi McFarland.

**Validation:** Kinley Kinley, Tashi Tobgay.

**Visualization:** Lekey Khandu.

**Writing – original draft:** Vinita Saxena.

**Writing – review & editing:** Audrey Xu, Kinley Kinley, Tashi Tsheten, Tenzin Gyeltshen, Tashi Tobgay, Tae Young Zajkowski, Willi McFarland, Lekey Khandu.

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
