## [Decision Letter · Decision Letter 0]

20 Apr 2023

PONE-D-23-07072Experiences of Stigma and Discrimination Against Transgender Men in BhutanPLOS ONE

Dear Dr. Saxena,

Thank you for submitting your manuscript to PLOS ONE. After careful consideration, we feel that it has merit but does not fully meet PLOS ONE’s publication criteria as it currently stands. Therefore, we invite you to submit a revised version of the manuscript that addresses the points raised during the review process.

We look forward to receiving your revised manuscript.

Kind regards,

Ricardo de Mattos Russo Rafael, Ph.D.

Academic Editor

PLOS ONE

Journal Requirements:

Additional Editor Comments (if provided):

Dear authors,

Firstly, congratulations on your manuscript. After a thorough analysis of the material, the reviewers have provided important and commendable feedback on the study. However, despite the excellent execution mentioned by the reviewers, the final decision is "minor revision".

For the publication of this manuscript, it is essential to clarify the theoretical and conceptual aspects indicated in the reviewers' evaluation and, above all, to complement the description of the control variables used in the multiple regression. There is still doubt about the measurement of the variable "stigma experience," which I consider essential to be clarified, as well as improvements in the text that will certainly add value to the study.

Even though I understand the difficulties in disclosing the database, it is essential that the authors review the guidelines of Plos One. In addition to not identifying the participants, in some cases, it is possible to hide potential variables that may reveal identities. However, note that it is not acceptable for the authors to be solely responsible for ensuring data access by the journal's guidelines.

Upon resubmitting your revised manuscript, please upload the minimal underlying data set of your study as Supporting Information files or to a stable, public repository, and include the relevant URLs, DOIs, or accession numbers in your revised cover letter. For a list of acceptable repositories, please refer to http://journals.plos.org/plosone/s/data-availability#loc-recommended-repositories. Any potentially identifying patient information must be fully anonymized.

Important: If there are ethical or legal restrictions to sharing your data publicly, please explain these restrictions in detail. Please refer to our guidelines for more information on what we consider unacceptable restrictions to publicly sharing data: http://journals.plos.org/plosone/s/data-availability#loc-unacceptable-data-access-restrictions. I reiterate that it is not acceptable for the authors to be solely responsible for ensuring data access. Therefore, if it is impossible to meet the above guidelines, it is essential that the authors fulfill this request.

Best Regards,

Ricardo de Mattos Russo Rafael, PhD

Academic Editor

PLOS ONE

Reviewers' comments:

Reviewer's Responses to Questions

**Comments to the Author**

1. Is the manuscript technically sound, and do the data support the conclusions?

Reviewer #1: Yes

Reviewer #2: Yes

2. Has the statistical analysis been performed appropriately and rigorously? 

Reviewer #1: Yes

Reviewer #2: Yes

3. Have the authors made all data underlying the findings in their manuscript fully available?

Reviewer #1: No

Reviewer #2: No

4. Is the manuscript presented in an intelligible fashion and written in standard English?

Reviewer #1: Yes

Reviewer #2: Yes

5. Review Comments to the Author

Reviewer #1: Thank you for the opportunity to review your manuscript. This highly important and well-conducted study addresses an understudied population: Asian transgender men. I have just a few suggestions aimed at improving the manuscript’s quality.

INTRODUCTION: Is it possible to include one brief paragraph describing the reality for LGBTQ+ persons in Bhutan? Is it legal to be openly queer? Does Bhutan recognize same-sex marriage? LGBTQ+ people can adopt kids, have legal recognition as a couple etc? Do trans persons have easy access to hormone therapy, sex reassignment surgery, changing name/gender in official documents etc.?

METHODS, measures: Why did the authors dichotomize ‘stigma experience’ and ‘discrimination’ in a different way (“often” vs. “sometimes/never” // “often/sometimes” vs. “never”)

RESULTS: what does it mean to have an occupation in “entertainment”? Does it include sex work?

RESULTS, table 4: Multivariate analysis was controlled by which variables?

DISCUSSION, first paragraph: There is no need to cite (again) your results.

DISCUSSION, first paragraph: “Our data corroborate high levels of stigma and discrimination suggested by the few available studies of trans men worldwide.” Please cite those studies after this sentence.

DISCUSSION, first paragraph: "Respondents specifically mentioned (…)” Respondents of which study? Please cite again.

DISCUSSION, first paragraph: “At least one quantitative study, from Peru, reported higher levels of discrimination experienced by healthcare providers (…)” The sentence is confusing; this Peruvian study was conducted with transgender men who were healthcare providers?

DISCUSSION, second paragraph: This sentence is also confusing, I’m not sure what the authors wanted to state here: “The finding that trans men who were students or unemployed did not report higher discrimination was unexpected as discrimination from academic and employment opportunities have been previously noted in studies of transgender persons.”

I look forward to seeing the manuscript published!

Reviewer #2: The manuscript presents the results of a pioneering study conducted among transgender men in Bhutan. All methodological procedures are appropriate for the population and sample. However, there is a conceptual imprecision, as the terms "stigma" and "discrimination" are used as categorical variables without clear definitions. The text does not specify what the authors mean when they ask "have you ever experienced stigma" - is this about experiences in general, as a transgender man? Additionally, in the next question, the term "experience discrimination" is used in the context of health services, which is clearer to readers than "experience stigma." The latter is typically used in passive voice to describe a feeling or perception of being stigmatized. Therefore, I kindly urge the authors to provide proper definitions and contextualization for these terms. Any translation issues should also be addressed.

6. PLOS authors have the option to publish the peer review history of their article (what does this mean?). If published, this will include your full peer review and any attached files.

Reviewer #1: **Yes: **Monica Malta

Reviewer #2: **Yes: **Helena Maria Scherlowski Leal David

---

## [Author Response · Author response to Decision Letter 0]

1 Jun 2023

Dear Dr. Ricardo de Mattos Russo Rafael,

We are thrilled that you and the reviewers found merit in our study with trans men in Bhutan and that PLOS ONE will consider the publication, pending satisfactory revisions. We have carefully considered all of the reviewers’ and editor’s comments and have done our best to address them. The following is a point-by-point response to the issues raised and revisions to the manuscript. 

Dear authors,

Firstly, congratulations on your manuscript. After a thorough analysis of the material, the reviewers have provided important and commendable feedback on the study. However, despite the excellent execution mentioned by the reviewers, the final decision is "minor revision".

Response: Thank you for the encouraging words and the decision of “minor revision”. 

For the publication of this manuscript, it is essential to clarify the theoretical and conceptual aspects indicated in the reviewers' evaluation and, above all, to complement the description of the control variables used in the multiple regression. There is still doubt about the measurement of the variable "stigma experience," which I consider essential to be clarified, as well as improvements in the text that will certainly add value to the study.

Response: We appreciate the comment and the complexity of measuring stigma. Added complexities include how such measures are translated into Dzongkha and interpreted by respondents. This area is well worth primary, qualitative, theoretical research in the context of Bhutan. However, as a rapid survey that was the first of its kind, our measure should be considered a broad indicator that aligns with self-perceived/evaluated stigma. We have clarified in the text of the revised manuscript in the Methods section providing the translation into English from Dzongkha. We also add the related limitations to the Discussion section and the need for more in-depth, contextual research. We point to references in the literature stigma frameworks with which we feel our indicator aligns. 

With respect to the control variables in the multivariate regression, please see our comments below in response to reviewer #1. 

Even though I understand the difficulties in disclosing the database, it is essential that the authors review the guidelines of Plos One. In addition to not identifying the participants, in some cases, it is possible to hide potential variables that may reveal identities. However, note that it is not acceptable for the authors to be solely responsible for ensuring data access by the journal's guidelines. Upon resubmitting your revised manuscript, please upload the minimal underlying data set of your study as Supporting Information files or to a stable, public repository, and include the relevant URLs, DOIs, or accession numbers in your revised cover letter. For a list of acceptable repositories, please refer to http://journals.plos.org/plosone/s/data-availability#loc-recommended-repositories. Any potentially identifying patient information must be fully anonymized. Important: If there are ethical or legal restrictions to sharing your data publicly, please explain these restrictions in detail. Please refer to our guidelines for more information on what we consider unacceptable restrictions to publicly sharing data: http://journals.plos.org/plosone/s/data-availability#loc-unacceptable-data-access-restrictions. I reiterate that it is not acceptable for the authors to be solely responsible for ensuring data access. Therefore, if it is impossible to meet the above guidelines, it is essential that the authors fulfill this request.

Response: We have re-read the policy; we agree with the principles of openly verifiable data that is not the decision of the investigators. Unfortunately, the protocol approved by Bhutan’s IRB explicitly says only the investigators or their designated supervisees are permitted access to the full data. In addition, the approved informed consent explicitly indicated who would have access to the data and this was what participants agreed to. For readers who want to know the numbers behind means, medians, and associations, we are available to provide those statistics and numbers without the possibility of identifying individuals. We appreciate that PLOS ONE prioritizes the safety of research participants and the ethical guidelines of the country of the data’s origin. 

Reviewers' comments:

Reviewer's Responses to Questions

Comments to the Author

1. Is the manuscript technically sound, and do the data support the conclusions?

Reviewer #1: Yes

Reviewer #2: Yes

2. Has the statistical analysis been performed appropriately and rigorously?

Reviewer #1: Yes

Reviewer #2: Yes

3. Have the authors made all data underlying the findings in their manuscript fully available?

Reviewer #1: No

Reviewer #2: No

Response: As noted above, we understand and do agree with the importance and principles of open data policies and that only rare exceptions are made. Please see the above response to the editor. 

4. Is the manuscript presented in an intelligible fashion and written in standard English?

Reviewer #1: Yes

Reviewer #2: Yes

5. Review Comments to the Author

Reviewer #1: Thank you for the opportunity to review your manuscript. This highly important and well-conducted study addresses an understudied population: Asian transgender men. I have just a few suggestions aimed at improving the manuscript’s quality.

Response: We are glad that you recognize the importance of this population and that the health needs of trans men are understudied. 

INTRODUCTION: Is it possible to include one brief paragraph describing the reality for LGBTQ+ persons in Bhutan? Is it legal to be openly queer? Does Bhutan recognize same-sex marriage? LGBTQ+ people can adopt kids, have legal recognition as a couple etc? Do trans persons have easy access to hormone therapy, sex reassignment surgery, changing name/gender in official documents etc.?

Response: We have updated the background text to include the requested context. According the UNAIDS and Bhutan’s legal codes, there is no criminalization of transgender people, sex work, or same-sex sexual acts in private [11]. Bhutan’s constitution protects transgender persons for acts such as rape, domestic violence, and sexual harassment in the workplace regardless of gender. However, laws do not explicitly protect LGBTQ+ individuals from discrimination on the basis of sexual orientation, gender identity, and gender expression. In addition, there are no explicit legal rights for transgender persons to access gender-affirming care, marry, or change government identity documents. Pride Bhutan and Queer Voices of Bhutan, two civil society organizations advocating for LGBTQ+ rights, are not legally registered organizations [12]. It should be recognized that the legal and cultural situation in Bhutan is rapidly changing [ref]. 

METHODS, measures: Why did the authors dichotomize ‘stigma experience’ and ‘discrimination’ in a different way (“often” vs. “sometimes/never” // “often/sometimes” vs. “never”)

Response: We clarify in the revised text the reason for these different dichotomizations. The different dichotomizations were based on the different distributions of these variables. “Often” reporting stigma was common, while any stigma experience was reported by greater than 95% of trans men, therefore there were too few to model correlations with reporting “any” vs. “never”. In contrast, the frequency of “often” experiencing discrimination was rare and too infrequent to model, while any experience (often or sometimes) was more balanced and therefore permitted modeling correlations. 

RESULTS: what does it mean to have an occupation in “entertainment”? Does it include sex work?

Response: We added clarification of “entertainment” as an occupation in the revised Methods. The occupations labeled as “entertainment” included employment in bars, karaoke clubs, and “drayangs” where dances are performed for customers [16]. These occupations do not explicitly include sex work, although some persons working in such venues might engage in sex work or transactional sex on the side. In the study, few (4.0%) trans men worked in such venues. Although we note that bivariate analysis indicated this type of employment was apparently associated with a higher experience of stigma, the finding did not hold in multivariate analysis.

RESULTS, table 4: Multivariate analysis was controlled by which variables?

Response: We have clarified in the revised Methods and Table 4 which variables were controlled for. Namely, the adjusted ORs are controlled for by the other variables listed for the models. Notably, age, education, and other demographic variables were not associated with stigma or discrimination, nor did they confound the other associations included in the final models. The revised Methods section further clarifies the selection of variables for inclusion in the final multivariate analysis.

DISCUSSION, first paragraph: There is no need to cite (again) your results.

Response: Thank you, the redundancy has been eliminated. 

DISCUSSION, first paragraph: “Our data corroborate high levels of stigma and discrimination suggested by the few available studies of trans men worldwide.” Please cite those studies after this sentence. 

Response: We appreciate your attention to detail. These studies have been added (citations 10-13,17)

DISCUSSION, first paragraph: "Respondents specifically mentioned (…)” Respondents of which study? Please cite again.

Response: Thank you, this unclear detail has been corrected. Citation 11 has been added to the statement. 

DISCUSSION, first paragraph: “At least one quantitative study, from Peru, reported higher levels of discrimination experienced by healthcare providers (…)” The sentence is confusing; this Peruvian study was conducted with transgender men who were healthcare providers?

Response: This was discrimination faced by trans men from healthcare workers. This has been corrected. 

DISCUSSION, second paragraph: This sentence is also confusing, I’m not sure what the authors wanted to state here: “The finding that trans men who were students or unemployed did not report higher discrimination was unexpected as discrimination from academic and employment opportunities have been previously noted in studies of transgender persons.”

Response: Thank you, we have rewritten the sentence to be clearer. 

I look forward to seeing the manuscript published!

Response: We agree!

Reviewer #2: The manuscript presents the results of a pioneering study conducted among transgender men in Bhutan. All methodological procedures are appropriate for the population and sample. 

Response: We appreciate the recognition of the importance of the study and the appropriateness of the methods for this population and context.

However, there is a conceptual imprecision, as the terms "stigma" and "discrimination" are used as categorical variables without clear definitions. The text does not specify what the authors mean when they ask "have you ever experienced stigma" - is this about experiences in general, as a transgender man? Additionally, in the next question, the term "experience discrimination" is used in the context of health services, which is clearer to readers than "experience stigma." The latter is typically used in passive voice to describe a feeling or perception of being stigmatized. Therefore, I kindly urge the authors to provide proper definitions and contextualization for these terms. Any translation issues should also be addressed.

Response: We agree with these distinctions and have revised the text accordingly:

Translated from the Dzongkha language, the question about stigma was generally asking about experience as a trans man, namely, “Have you experienced stigma because people knew or thought you are a trans man?” The question about discrimination referred to experience with accessing healthcare, namely, “Have you experienced discrimination when accessing health services because people knew or thought you are a trans man?” 

As noted above and in the limitations, our measures of these perceptions and experiences are brief indicators and we concur with the need for primary theoretical research in this area. 

We have also carefully reviewed the journal requirements and formatting as directed. In the course of revision, we also identified and corrected minor typos. With these changes, we feel the manuscript has been greatly improved. We thank the reviewers for their time and attention and hope for a favorable response from PLOS ONE. 

Sincerely, 

Vinita Saxena on behalf of the authors

---

## [Decision Letter · Decision Letter 1]

13 Jun 2023

Stigma and Discrimination Against Transgender Men in Bhutan

PONE-D-23-07072R1

Dear Dr. Saxena,

We’re pleased to inform you that your manuscript has been judged scientifically suitable for publication and will be formally accepted for publication once it meets all outstanding technical requirements.

Kind regards,

Ricardo de Mattos Russo Rafael, Ph.D.

Academic Editor

PLOS ONE

Reviewers' comments:

Reviewer's Responses to Questions

**Comments to the Author**

1. If the authors have adequately addressed your comments raised in a previous round of review and you feel that this manuscript is now acceptable for publication, you may indicate that here to bypass the “Comments to the Author” section, enter your conflict of interest statement in the “Confidential to Editor” section, and submit your "Accept" recommendation.

Reviewer #1: All comments have been addressed

Reviewer #2: All comments have been addressed

2. Is the manuscript technically sound, and do the data support the conclusions?

Reviewer #1: Yes

Reviewer #2: Yes

3. Has the statistical analysis been performed appropriately and rigorously? 

Reviewer #1: Yes

Reviewer #2: Yes

4. Have the authors made all data underlying the findings in their manuscript fully available?

Reviewer #1: No

Reviewer #2: Yes

5. Is the manuscript presented in an intelligible fashion and written in standard English?

Reviewer #1: Yes

Reviewer #2: Yes

6. Review Comments to the Author

Reviewer #1: (No Response)

Reviewer #2: All the conceptual issues and imprecisions were adequately addressed by the authors, and the manuscript is ready to be published.

7. PLOS authors have the option to publish the peer review history of their article (what does this mean?). If published, this will include your full peer review and any attached files.

Reviewer #1: **Yes: **Monica Malta

Reviewer #2: **Yes: **Helena Maria Scherlowski Leal David

---

## [Editor Report · Acceptance letter]

12 Jul 2023

PONE-D-23-07072R1 

Stigma and Discrimination Against Transgender Men in Bhutan 

Dear Dr. Saxena:

I'm pleased to inform you that your manuscript has been deemed suitable for publication in PLOS ONE. Congratulations! Your manuscript is now with our production department. 

Kind regards, 

on behalf of

Dr. Ricardo de Mattos Russo Rafael 

Academic Editor

PLOS ONE